# Healthcare Professionals’ Experiences and Perspectives of Facilitating Self-Management Support for Patients with Low-Risk Localized Prostate Cancer via mHealth and Health Coaching

**DOI:** 10.3390/ijerph20010346

**Published:** 2022-12-26

**Authors:** Louise Faurholt Obro, Palle Jörn Sloth Osther, Jette Ammentorp, Gitte Thybo Pihl, Peter Gall Krogh, Charlotte Handberg

**Affiliations:** 1Urological Research Center, Vejle Hospital—Part of Lillebaelt Hospital, University Hospital of Southern Denmark, 7100 Vejle, Denmark; 2Department of Regional Health Research, Faculty of Health Sciences, University of Southern Denmark, 5000 Odense, Denmark; 3Centre for Research in Patient Communication, Odense University Hospital, 5000 Odense, Denmark; 4Department of Clinical Research, University of Southern Denmark, 5000 Odense, Denmark; 5UCL University College, 7100 Vejle, Denmark; 6Department of Engineering, Aarhus University, 8200 Aarhus, Denmark; 7The National Rehabilitation Center for Neuromuscular Diseases, 8000 Aarhus, Denmark; 8Department of Public Health, Faculty of Health, Aarhus University, 8000 Aarhus, Denmark

**Keywords:** prostate cancer, self-management, well-being, mHealth, health coaching, interpretive description, self-determination theory

## Abstract

**Introduction:** Self-monitoring of self-management interventions with the use of mobile health (mHealth) can enhance patients’ well-being. Research indicates that mHealth and health coaching act symbiotically to providing a more constructive outcome. Nurse coaches seem to have a significant role in translating the patients’ tracked data. **Objective:** The objective was to explore healthcare professionals’ experiences of an intervention offering self-management support through mHealth and health coaching for patients with prostate cancer. **Methods:** We used the interpretive description methodology, combining semi-structured individual and focus group interviews and participant observations of patient-coach interactions and use of mHealth in coaching sessions. The study was conducted between June 2017 and August 2020. **Results:** The nurse coaches experienced motivation and autonomy when possessing the right competences for coaching. Furthermore, the nurse coaches experienced conflicting expectations of their roles when having to integrate mHealth. **Conclusion:** The experience of being competent, autonomous, and confident is important for the nurse coaches to be mentally present during the coaching sessions. On the other hand, the findings indicate that having the sense of not being confident in one’s own ability to perform leads to reduced motivation.

## 1. Introduction

### 1.1. Background

The incidence of low-risk prostate cancer (PCa) has increased over the past two decades [1,2,3], and in this population active surveillance (AS) and watchful waiting have been shown to be feasible management strategies [2,4]. In addition, quality of life has been demonstrated to be higher in men undergoing AS than in men receiving active treatment [2,5]. On the other hand, research suggests that patients with PCa under AS experience higher rates of anxiety and depression than other men [6], which was reflected in the PRIAS study, revealing that ~13% of men on AS chose to initiate active treatment for PCa due to anxiety [7].

Anxiety driven surgical and medical interventions result in unnecessary morbidity and inappropriate use of resources. Studies have indicated that healthy lifestyle interventions may reduce the number of patients with low-risk PCa undergoing active treatment and seem to modulate the biological processes involved in tumor progression [8,9,10,11]. Additionally, it has been shown that a healthier lifestyle and physical activity increase well-being and quality of life in men with PCa [8,9,10,11]. To empower patients to manage their own health, lifestyle, and well-being, self-management (SM) strategies seem to be effective tools, and SM is becoming an important paradigm for healthcare professionals [12,13,14,15]. Meta-analyses suggest that SM strategies improve health outcome as well as psychosocial and behavioral outcomes [16]. An important component of self-management is self-monitoring, where a patient monitors, e.g., weight and blood pressure or symptoms like pain and fatigue [17,18].

Self-monitoring of SM interventions with the use of mobile health (mHealth) engages and empowers patients in their own health and well-being [19]. Furthermore, mHealth offers opportunities for healthcare professionals to extend patient care efforts outside the care setting, thereby supporting SM [20]. mHealth can be seen as the use of wireless technologies and wearables (e.g., mobile phones and self-monitoring devices) to monitor and support the achievement of health objectives [21]. Through continuous self-monitoring, patients become more aware how daily decisions and behavior may affect health and well-being [19,20,21]. mHealth also offers healthcare professionals the ability to evaluate a prescribed course of action, monitor adverse events, and identify areas for improvement [19,20,21]. Although the advantages of mHealth seems persuasive, some challenges have been pointed out [21,22]. A considerable challenge is that many patients have barriers to adopting mHealth in everyday-life [22]. Therefore, Woods and colleagues recommend that healthcare professionals support the patients to a greater extent as a part of the SM strategy [22]. A scoping review indicated that mHealth and health coaching act symbiotically to enhance patients’ SM [23], providing a more constructive outcome [24,25,26]. Health-coaching is seen as health education in a coaching setting, with the aim of enhancing well-being and to facilitate the achievement of health-related goals [26]. In addition, we previously found that nurse coaches seem to play a significant role in translating the patient’s tracked data into advice for daily living [27]. Since the role of the nurse coaches seems to be an essential part of this strategy, investigation of their experiences and perspectives of this new approach may qualify the intervention.

### 1.2. Objective

To explore the healthcare professionals’ experiences and perspectives of an intervention offering self-management support through mHealth and health coaching for patients with PCa in surveillance management.

## 2. Method

### 2.1. Study Design

We used the interpretive description (ID) methodology to explore healthcare professionals’ experiences and perspectives of the intervention, by combining semi-structured individual and focus group interviews and participant observations of patient-coach interactions and use of mHealth in coaching sessions. ID is an inductive research strategy designed to explore questions that arise from clinical practice and to generate practical knowledge [28]. To capture the influence of mHealth and health coaching on healthcare professionals’ psychological needs and motivation, we applied the self-determination theory (SDT) [29]. According to the SDT, people are intrinsically or extrinsically motivated to perform a certain task, which depends on the degree to which their universal psychological needs for autonomy, competence, and relatedness are fulfilled [29].

### 2.2. Setting and Sampling

Convenience sampling was used to select four urological nurses to coach the patients [28]. The only criteria for inclusion were that the participating nurse coaches worked in the urological outpatient clinic. The nurses were included between June 2017 and August 2020; they were all females aged between 30 and 52 years old. The nurse coaches participated in two individual interviews and one focus group interview. A medical doctor responsible for the patients was also recruited for the focus group. All participants were recruited from the Urological Department at Vejle Hospital—part of Lillebaelt Hospital, Vejle, Denmark.

### 2.3. Training of the Nurses

Group 1

Group 1 consisted of two nurses; they completed a 1-day health coaching course. The course focused on a general introduction to health coaching, and the nurses were coached by a health coach. The health coach had developed the content of the training based on her experiences with teaching healthcare professionals to coach patients. After having coached a few patients, the nurses requested more specific knowledge on health coaching, e.g., questioning techniques and tools to facilitate a coaching session.

Group 2

Based on the feedback from the nurses in group 1, the course was extended by 1 day. The last two nurses were recruited in group 2. They completed a 2-day course. The first day was similar to that offered in the course completed by group 1. On the second day, the nurses were introduced to the coaching tool “Wheel of Life” [30] (Figure 1a) and coaching questioning techniques. The “Wheel of Life” is a recognized tool in order to help structure a coaching conversation [30]. The tool was chosen by the health coach. Both courses were conducted by a certified health coach.

### 2.4. Intervention

The coaching program aimed at providing ongoing support and guidance for the patients to set sustainable goals and achieve goals and life-style changes that improved overall health and well-being. The nurse coaches primarily addressed goals related to the participants’ physical activity, healthy eating behavior, or both. In addition, the nurse coaches facilitated conversations to address the participants’ strengths, motivation to change, and perceived barriers.

The nurse coaches in group 1 and the patients identified areas needing improvement. All patients identified water intake as an area for improvement and agreed to register daily water intake with a Bluetooth button (Bttn^®^, Amsterdam, The Netherlands) (Figure 2a) connected to an app “My Course”, which is an app that is connected to the patients’ electronic patient journal. The patients could see their tracked data in the app. The patients’ were asked to give the Bluetooth button a push, every time they drank a glass of water. At the following sessions, the coach and the patients evaluated the self-tracked data and talked about barriers and motivation for achieving the goal of increased water intake.

The nurse coaches in group 2 used the Wheel of Life [30] to identify and set goals in cooperation with the patients. The Wheel of Life [30] is a tool assessing where coaching may be beneficial (Figure 1a). Each area on the wheel was defined by the patient and rated from 1 (not at all satisfied) to 10 (completely satisfied/could not be improved) by drawing a line across the corresponding number in that section of the pie and then shading below it. Subsequently, the patient and the coach were left with a jagged wheel that illustrated areas for growth (see Figure 1b for an example of a completed version). In order to improve the patients’ overall health and well-being, the patients were equipped with two mHealth devices: a Smart-tracker (FitBit 4 (Fitbit©., San Francisco, CA, USA) or Garmin vívosport^®^ (Olathe, KS, USA)) and a music device (Figure 2b). The objective of the use of a Smart- tracker for the patients was to raise their daily activity and by tracking their steps. At the coaching sessions, the coach-nurses and patients would talk about how the patients felt after increasing their daily activity. Additionally, the patients would be asked to address barriers and facilitators related to be more active. In order to improve the patients’ well-being, the patients were asked to reflect on own life and emotions. To facilitate the patients’ self-reflection a music device was developed. The rational for using a music device were based on findings from a small pilot study, which we conducted before conducting the present study. The aim of the pilot study was to explore what men with PCa valued in their daily living, and what they associated with own well-being. We found that some men experienced well-being when they listened to music, and that the men often talked about the music as a language of emotions Therefore, we developed a music device together with an engineer from Department of Computer Science, Aarhus University. The music device was intended to function as a digital diary where patients were asked to play a melody on the music device that reflected their well-being, feelings and emotions on a daily basis. The coach and patient would listen to the recordings at the coaching sessions as a basis for their conversation on the patients’ well-being.

In total, the nurse-coaches coached 13 patients for 19 weeks. The 19 weeks program included eight individual coaching-sessions: four face-to-face visits in the outpatient clinic, lasting between 45 and 60 min each, and four telephone calls of 30 min each (Figure 3). More information regarding the intervention is reported elsewhere [27].

### 2.5. Data Generation

The first author conducted eight semi-structured individual interviews with the nurse coaches: four interviews before the intervention and four at the end of study. The nurse coaches were interviewed (30–70 min) in the clinic. The focus group interview lasted 70 min. The focus group interviews were conducted to explore the meanings, beliefs, and cultures that influence healthcare professionals’ feelings, attitudes, and behaviors [31]. The interviews were audiotaped and transcribed verbatim. To detect the impact of contextual events impacting the study [28], observations were carried out in October 2017 and October 2020 (approximately 5 h of observations). Data from the observations were used to provide insight into coach and participant interaction and to support and elaborate interview data.

### 2.6. Data Analysis

The analysis was guided by the ID methodology [28]. The interview data were transcribed, anonymized, and transferred to the software program QRS NVivo 12 ©. In the first step of the analysis, the first author LFO and the last author CH read the transcripts and developed an initial coding structure based on the initial analysis. Then, LFO and CH performed a more specific coding strategy, shifting between the process of coding and taking “a step back” to gain a perspective of the data material as a whole. After the initial coding, LFO and CH further refined, described, and discussed themes grounded in the remaining data, and this step was repeated, ensuring that the themes comprised the data, and subsequently, the rest of the research team (PJSO, PGK, JA and GTP) was involved in further discussions. Correspondingly, we addressed any inconsistencies both within and between the interview data, field notes, and notes from the coaches. Finally, we created a model representing the analytical findings in a hierarchy [28].

### 2.7. Ethical Considerations

The study was approved by the Regional Committees on Health Research Ethics for Southern Denmark (case ID: 20212000-105). All participants received written and oral information about the study and its purpose. We obtained informed consent, and the participants were informed that participation was voluntary. All data were anonymized and stored in a secure place approved by the Danish Data Protection Agency and in accordance with the General Data Protection Regulation (GDPR).

## 3. Results

Analysis revealed three overarching and interacting themes (Figure 4). The first theme, *The influence of coaching*, indicated that the nurse coaches experienced motivation and autonomy when possessing the right competencies for coaching. This helped the nurses to sustain mentally presence and be focused and engaged in the here and now. The second theme, *The influence of mHealth*, revealed that the nurses struggled with staying motivated due to lack of competences related to the mHealth devices. Furthermore, the nurse coaches experienced diverse expectations and an uncertainty of their roles when having to integrate mHealth as a part of the coaching sessions. The third and final theme, *Relations through shared experiences*, indicated that the nurses experienced the interaction between them and the patients as important and that this interaction led to a unique relationship (Figure 4).

The nurse coach’s ability to be mentally present with the patients in the coaching sessions seemed to influence their experiences and perspectives of the intervention. The nurse coaches indicated during the interviews that they found themselves in a dynamic continuum of being mentally present and/or being mentally absent in the coaching session, with the coaching being more beneficial and meaningful for the patients when the nurse coach managed to stay mentally present.

### 3.1. The Influence of Coaching

The nurse coaches described an experience of being mentally present when they were coaching the patients. Some of the nurses explained that they could be more mentally present during a coaching session than during a traditional consultation in the outpatient clinic because they found coaching more suitable for surveillance management. A nurse explained:


*“This group of PCa patients requires a different approach than other groups of patients; they need emotional support and a talk about living with an untreated cancer, rather than a blood test and a talk about their medicine.”*
(Nurse coach #2)

In the focus group, one of the participants described that surveillance management was more psychological and existential than management of patients in active treatment. It focused more on making the patients feel safe and providing them with health advice. Due to the existential approach, they described that they experienced coaching to fit the management of these patients better than traditional out-patient consultations. During the focus group interviews, some of the healthcare professionals also described that the feeling of being mentally present was due to the coaching methods, which helped incorporate the patient’s individual needs and experiences:


*“I felt more mentally present with the patients. With coaching, I was able to find out what was important for the patients to talk about. I think that the coaching tools made it possible for me to accommodate their emotional needs and to open up for the existential talks.”*
(Nurse coach #3)

The nurse coaches experienced that coaching gave them techniques to include the patient’s individual and existential needs in their management, and this led to a feeling of being mentally present with the patients.

### 3.2. Motivation through Embedded Competences for Coaching

Some of the nurse coaches described that they felt that they had coaching competences and were motivated to use this approach. They explained they had signed up for the project because they felt skilled in communication, and saw it as an important part of the nursing profession. Other nurses were motivated to coach the patients because they experienced that the patients needed someone to talk to. A nurse explained:


*“I think that coaching can be used to help the patients through their treatment. To get someone to talk to. Allowing them to open up and finding their needs.”*
(Nurse coach #1)

Likewise, in one of the last coaching sessions a patient expressed great satisfaction with being in the intervention and told the nurse that she had been a good listener. The nurse expressed gratitude for the patient’s reaction. Furthermore, several nurses experienced that coaching felt natural for them, as they described that communication was already a great part of their nursing tasks in the clinic and moreover was important in nurse education. Another nurse described motivation for coaching because coaching was an approach to let the patients become more active during the conversation. The nurse explained:


*“During a coaching conversation, I don’t have to come up with all the answers. I have to let the patient find the solution that is right for him. I don’t normally have this role in the outpatient clinic.”*
(Nurse coach #2)

The nurses indicated that they experienced that the right coaching education and training were important to their becoming motivated to coach. Some of the nurses explained that they felt that there was a lack of focus on questioning techniques during the coaching education. Whereas others explained having focused on questioning techniques during their coaching education. Some of the nurses expressed they regarded it as important to receive specific advice and teaching on how to pose coaching questions as they could not rely on their experience from their practice in the outpatient clinic.

From the observations and interviews, it appeared that when the nurses experienced themselves competent to coach and when they experienced that it was important for the patients, they were motivation. In addition, it seemed important for the nurses to have received the appropriate education and training to be motivated to coach.

### 3.3. Autonomy through Coaching

Some nurses experienced great satisfaction by having the ability to have more time with the patients when coaching compared with their tasks in the outpatient clinic. One nurse explained that:


*“…by being a coach, I am there for the patient, whereas as a nurse in the clinic, I sometimes feel like I am mainly there for the doctor.”*
(Nurse coach #1)

Autonomy seemed to motivate the nurses, who explained that their role as a coach was more autonomous than their traditional role, which is more like a “helping hand” or translator of what the doctor said. During coaching, a nurse explained that she felt more independent and that her new role gave her more job satisfaction. Similarly, another nurse said:


*“Coaching is something that nurses are instructed in, not just giving messages from the doctor to the patients.”*
(Nurse coach #1)

Another participant explained how coaching could become a new important and professionally defining task for nurses. The participant explained that coaching could give the possibility both for the healthcare professionals and the patients to identify what is important for the patients, and which choices that are best suited for the patients. It seemed that the nurses experienced that coaching gave them more dedicated time with the patient, which they experienced as motivating. Additionally, the nurses experienced coaching as a self-directed task that suited the nursing profession, in which they experienced autonomy.

### 3.4. The Influence of mHealth

The nurses articulated their experiences in a spectrum from motivated to less motivated as a consequence of having to integrate mHealth as a part of the coaching. They described that they were only motivated to use mHealth if it could benefit the coaching conversations and the patients and if they felt that they had the right competencies for using the mHealth device. Some nurses explained that the coaching conversation sometimes could be difficult to start because addressing emotions for some patients was abstract and problematic. However, the nurses described that the tracked data served as a tool to start the conversation; a nurse explained:


*“In one of the first sessions we talked a lot about his water intake, but in the last session the conversation was quite different. We talked a lot about his relationship with his wife and that they never talked about his disease, which he was depressed about.”*
(Nurse coach #1)

Talking about more specific things, like activity, often led to more emotional subjects being discussed in subsequent sessions. Nevertheless, some nurses experienced not having had enough training in the use of the mHealth devices, which occasionally caused a feeling of being mentally absent during the coaching sessions. During an observation of a coaching session, a nurse appeared to be mentally absent when she was trying to make one of the patient’s mHealth devices function. The nurse tried to start it and focused on the device when the patient told her about his voiding problem and that he had trouble sleeping at night. The patient was talking about how he felt exhausted during the daytime; however, it did not seem that the nurse was listening, and the conversation stopped. When the nurse placed the device on the table, the patient said nothing, and they both seemed to have lost focus. The problems with the mHealth devices seemed to negatively affect the nurses’ ability to be mentally present. Likewise, a nurse explained in an interview:


*“It was difficult to be present in the conversation when I had to think about how to get the device to work again.”*
(Nurse coach #2)

Moreover, some of the nurses described that the influence of technology also seemed to make the patients mentally absent during coaching. They explained that some patients were more focused on their data and the devices, and that it seemed like some of the patients did not understand that they were also there for a talk about the emotional aspects of their disease. It appeared that technology could deprive the nurses and patients of presence in the conversations. This aspect seems to affect the coaching negatively.

### 3.5. Lack of Motivation through Unintegrated Competences for mHealth

As indicated above, several nurses described that they did not feel suitably trained in the use of the mHealth devices, and therefore often experienced lack of technical competences related to mHealth, although all of the nurses were experienced users of different kinds of technologies, e.g., smartwatches, smartphones, smart trackers, and tablets, and they rated their competence level in technology from good to advanced. The nurses explained that they could not make some of the mHealth devices start after replacing the batteries, and they experienced concern about destroying the device accidentally. When the nurses experienced not having the right competence for the technology, they seemed less motivated to use the devices. A nurse described:


*“I was afraid of ruining it so the patient couldn’t use it again. I didn’t know these devices very well, so I did not want to do anything stupid. We agreed not to use it for the rest of the intervention.”*
(Nurse-coach #3)

In continuation, some of the nurses highlighted that it was important for them to have the opportunity to call for support if technical issues occurred. The lack of motivation also seemed to arise when the nurses experienced not having the competence to interpret the data they should use in the coaching sessions. This was observed during a coaching session, where the nurse and patient discussed how the patient’s data from his music devices should be interpreted. They laughed and said they did not know what to do with the tracked data. The nurse said in an interview that:


*“I found the music devices too abstract and too foolish. I couldn’t see how it would help the patients and I experienced that the patient felt the same.”*
(Nurse coach #1)

It seemed that the nurse’s motivation was affected negatively when they did not have the proper technical competences, and when they could not understand the aim and logic behind the music device. The feeling of not being motivated could in some cases end in rejection of the use of the devices.

### 3.6. Loss of Professionalism through Diverse Expectations of Roles

Some of the nurses experienced that they felt uncertain of their roles as coaches, which caused a feeling of loss of autonomy and not being professional. They explained that in the outpatient clinic they felt more in control and autonomous, having the knowledge and making the decisions, whereas being in the intervention had led to uncertainty of expectations regarding their own roles. The feeling of being unprofessional seemed to be linked to the nurse’s expectations of their role in the use of mHealth. A nurse explained:


*“You get out of your comfort zone, you don’t have control over this because you don’t know exactly how to handle the devices and how the patients will react, and the fact that you don’t know 100% what the outcome will be. It challenges you as a professional.”*
(Nurse-coach #4)

In a coaching session, a nurse coach (#3) had a feeling of not being in control and professional. The nurse and a patient were discussing how to interpret data from the patient’s music device. The nurse looked at her computer screen for a long time without talking to the patient. After several minutes, the nurse said that she did not know how the data should be interpreted and used in the session.

To some nurses, it seemed that their expectations regarding their own roles were also negatively affected by the patients’ expectations of the nurses’ roles in the intervention. Some nurses described that they experienced that the patients expected the nurses to be in control and knowledgeable regarding the applied mHealth technologies. A nurse (#2) wrote in her notes that a patient merely talked about data and the device and had a lot of questions about the device, which she could not answer, which frustrated him. Likewise, a nurse described that the patients seemed dismissive when they experienced that the nurses did not know how to operate a device and understand the data; the nurse explained:


*“I experienced that they were very dismissive of it. I also think that it was difficult to explain to them exactly what it was about because it was a bit abstract.”*
(Nurse coach #3)

It seemed to some nurses that having the right knowledge in relation to mHealth caused a feeling of being professional and in control. However, when the nurses experienced not having enough understanding of the devices, they experienced not being able to live up to their own expectations. Moreover, when the nurses found that the patients also expected that the nurses were in control and knowledgeable, they felt a stronger uncertainty of their own capacity in their roles in the intervention. The feeling of not being able to live up to their own and the patient’s expectations caused a feeling of loss of control and being unprofessional.

### 3.7. Relations through Shared Experiences

A mutual feeling of relations between the patients and the nurse coaches emerged during the intervention. The coaching sessions seemed to build a strong relationship between nurses and patients. The nurses described that the strong relationship motivated them and the patients to take part in the intervention. Some of the nurse coaches experienced that the conversations became more personal than they were in the outpatient clinic. A nurse coach explained that:


*“I experienced that coaching creates a more personal connection to the patient. A patient told me that he didn’t experience being just a number when he came for a coaching session. The patient said that in the outpatient clinic there was never time for anything else than giving information about his blood tests.”*
(Nurse coach #3)

The personal connection during the coaching sessions seemed to provide the nurses with the opportunity to sense if the patient had something they wanted to discuss but did not mention verbally. From the coach notes, it seemed that the personal connection motivated the nurses to explore and pursue the patient’s statements. A nurse wrote:


*“I could sense and hear that he said something between the lines. I asked him, and suddenly he opened up, and said that he had a hard time talking about emotions.”*
(Nurse coach #2)

The sense of relationship between the patients and nurse coaches also emerged using mHealth devices. Some of the nurses explained that when they and the patients experienced technical issues with the mHealth devices, it seemed to create a shared experience where both of them were novices in relation to the mHealth device. This occasionally catalyzed a more personal relationship. It appeared during a coaching session that a patient and the nurse were helping each other to make the device work properly. They exchanged experiences with technology and ended up laughing about it. Both the coaching and the use of mHealth seemed to create a strong relationship between the nurses and the patients in various ways. Their shared experiences through the intervention created sense of connectedness that again seemed to motivate the nurses and the patients.

## 4. Discussion

The nurses in the study found that coaching was an approach that was becoming more autonomous and paying more attention to what patients communicate, thereby creating a stronger relationship. The nurses trusted themselves to have the proper competencies for coaching, accumulated through their profession and the course, which led to task motivation. This finding is in accordance with the Self-determination Theory developed by Ryan and Deci [32], which argues that the feeling of being competent and autonomous is crucial for people’s satisfaction and motivation when new actions and tasks are introduced [32]. When the nurse coaches in the mentally present study experienced confidence in the tasks in the intervention, the tasks and actions became more meaningful. It appeared that their sense of presence was strengthened along with their ability to pay attention to what the patients tried to convey during the coaching session. Our findings are also consistent with Bandura’s social cognitive theory about self-efficacy. It highlights that people are enthused and motivated about a new task if they feel that they have the right competencies and are confident about their ability to succeed [33]. In contrast, if the nurses lacked confidence and the ability to operate the devices, this reduced their motivation to use the devices during coaching sessions. In a previous study, we found that the male patients were more likely to reject mHealth solutions if they did not have the right competencies and skills to use the devices [34]. Our findings also demonstrated that the nurse coaches’ sense of not having the proper technical competencies for mHealth could result in loss of autonomy and being unprofessional, further challenging the patients’ expectations of the nurse coaches’ roles. Research shows that to facilitate job satisfaction among nurses, autonomy and a sense of being professional increase motivation significantly [35]. One study pointed out that incongruence between nurses’ perceived role expectations and their achievement can be experienced as a loss of competencies, autonomy, and professionalism, resulting in the nurses being less motivated [27]. Other studies conclude that education, good working condition and ergonomic, supporting human factors and a clear consistent information about the behavior expected in a role are important for the nurses to feel confident and professional in their job [27,34,35,36]. This emphasizes the importance of providing education and support in both coaching and in the use of technology for the nurses to feel confident and professional when coaching and mHealth are introduced in clinical practice.

We also found that the devices fostered an interpersonal interaction and a strong relationship between the nurse coaches and the patients. The nurse coaches experienced an interpersonal interaction with the male patients, even when technical issues with the devices occurred because they shared the same problems. The nurse coaches found that this interaction led to a unique relationship catalyzing the nurse coaches to coach the patients. In line with this, studies have shown that a strong and trustful relationship between a patient and the nurse coach is crucial to facilitate coaching [26,37].

According to Reinhard Stelter, shared meaning-making and collaborative value reflections between the coach and the patient are the most central aspect of coaching [38]. Sharing the same experience, where the nurse-coaches and the male patients worked together to make the devices work, seemed to create a positive interaction and relationship, which were experienced as being fruitful for both the nurse and the patients. These findings are also supported by Hartmut Rosa, claiming that relations of resonance “lead to mutual reinforcement, thereby magnifying the amplitudes of the vibrations” by means of which agential elements can enter mutual fruitful modes of communication [39]. Furthermore, we found that the nurse coaches were motivated toward intervention due to the possibility of forming a strong relationship with patients. Several studies conclude that building strong relationships can be motivating, and that the sense of connectedness with others is a fundamental need for all humans [32,40].

### Study Strengths and Limitations

This study was conducted consistently within the framework of the ID methodology as outlined by Thorne [28]. The use of this framework ensured that the research was both methodologically and interpretively rigorous, both of which are required for qualitative research to be credible [40]. We enhanced the credibility through researcher triangulation. Our findings are strengthened by the author teams’ different backgrounds within healthcare and research, which provided various perspectives to the analysis and data interpretation. The analytical interpretations were strengthened through data triangulation, which also enhanced the credibility of our findings [28]. We used individual and focus group interviews, participant observations, and notes from the nurse coaches to help and challenge the interpretations of the nurse coaches’ experiences of the intervention. The diversity of the data generated gave us different insights into the intervention, strengthening the analytical interpretations [28]. Furthermore, the credibility was pursued through consistent analytic logic [28] with the same interview guide for all interviews and the four-step analytical process in accordance with ID, allowing the same structure and analytic consistency [28].

A limitation of our study may be that we only included four nurse coaches, and all of them were females. The small sample size may decrease the generalizability of our study [40]. It is likely that more healthcare participants and inclusion of male nurses could have further enhanced our understanding of the intervention. It is also not inconceivable that the gender composition, female nurse coaches and the male patients, could have affected the results. Moreover, owing to the explanation of the research context and the richness of data, our opinion is that the findings can be transferred to other healthcare professionals in other healthcare settings in Western countries [41]. Another limitation could be that we used convenience sampling, where only the nurses who wanted to participate were included. This sampling type may have resulted in a more positive experience of the nurse coaches, which might have impacted the findings. However, based on our data, we found that the nurses gave their honest experiences of the intervention and provided positive and negative experiences from the intervention. The difference in the training of the coaches could also have affected the findings. The nurses in group 2, who had most training, could have reflected more positive regarding the intervention than the nurses in group 1. Moreover, absence of education in the use of mHealth device could also have affected the finding in a negative direction, since the nurses’ motivation seemed to be related to having the right competences of the technology. Moreover, a limitation may be that the interviewer (first author) was known to the nurses, which could have led to more positive statements from the nurses. However, knowing the field is important to avoid any potential pitfalls, and being a part of the field can create trust among the nurses, enabling them to speak freely [28].

## 5. Conclusions

The experience of being competent, autonomous, and confident is significant for the nurse coaches to be mentally present during coaching sessions. On the other hand, the findings indicate that having the sense of not being confident in one’s own ability to perform a task, e.g., using the mHealth devices leads to reduced motivation and a sense of being less professional. Finally, the nurse coaches felt a strong interpersonal relationship with the patients when they experienced shared issues and problems with the mHealth devices. The shared meaning-making between the nurse coaches and the patients was motivational for the nurses and is essential for facilitating coaching in clinical practice. In conclusion, this study highlights that education and support are essential to ensure that nurse coaches feel confident and professional when introduced to new approaches in clinical practice.

To understand how these shared experiences using mHealth influence the relationship during a coaching intervention, future research should illuminate the perspective of how mHealth devices affect the nurse coach- male patient relationship. Future nurse coaching training should include a concept of developing strong relationship skills between the coaches and the patients. Finally, a consideration of the nurses’ digitally health literacy is necessary to provide education in mHealth.

## Figures and Tables

**Figure 1 ijerph-20-00346-f001:**
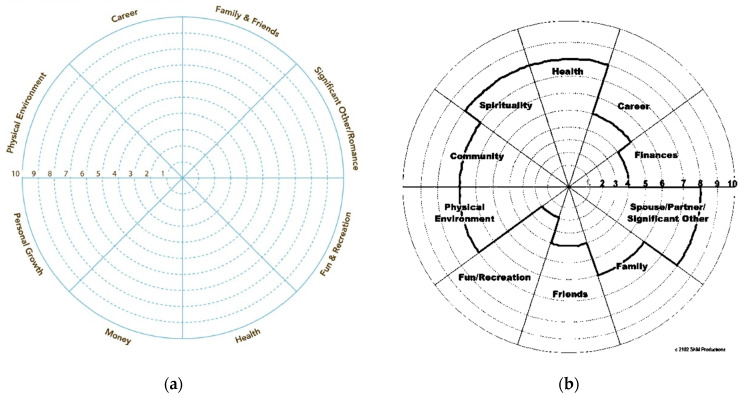
(**a**) Wheel of life by Paul J. Meyer; (**b**) Wheel of life by Paul J. Meyer (example completed).

**Figure 2 ijerph-20-00346-f002:**
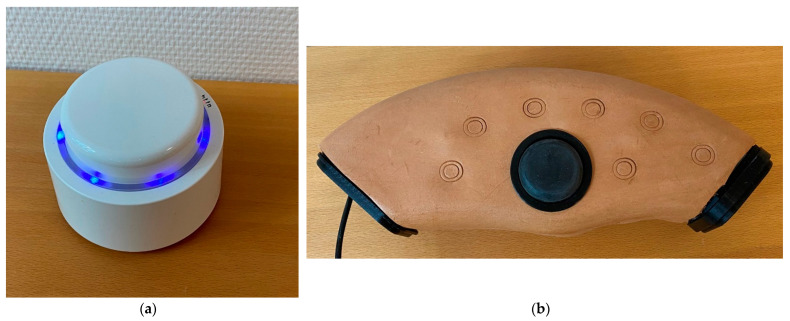
(**a**) Bluetooth button bt.tn; (**b**) Music device.

**Figure 3 ijerph-20-00346-f003:**
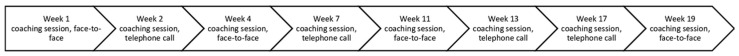
The 19-week coaching program.

**Figure 4 ijerph-20-00346-f004:**
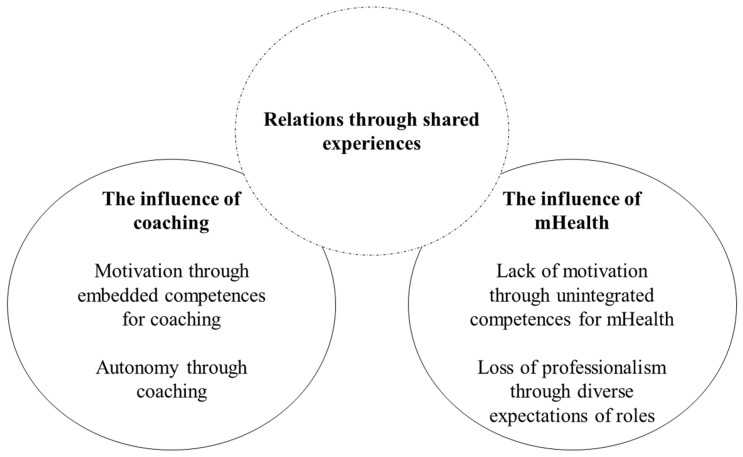
The three overarching and interacting themes.

## Data Availability

Data is unavailable due to privacy and ethical restrictions.

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
