# Peer review of "Healthcare Professionals’ Experiences and Perspectives of Facilitating Self-Management Support for Patients with Low-Risk Localized Prostate Cancer via mHealth and Health Coaching"

_ijerph, 2022, doi:10.3390/ijerph20010346_

Round 1

Reviewer 1 Report

Thank you for inviting me to you review the manuscript. The writing of this work is clear and succinct.

My overall feedback is sceptical for the  reasons: 

you had an extremely small participant cohort to report against 

there is no report/limitation regarding the difference in coaching training the nurses received, which may well have impacted the findings and I feel needs recognition in the limitation.  

You have provided no recommendations regarding

future coaching training

Technology literacy or future digital health education requirement

Author Response

Response to the Reviewers

Dear reviewers,

Thank you very much for your interest in our manuscript. We are grateful that you have decided to give us the opportunity to revise the manuscript according to the reviewer’s comments to meet your expectations for publication. We hereby resubmit our revised manuscript.

We appreciate the reviewers’ useful comments and the editorial instructions. We have answered all comments below point by point and followed the requested editorial instructions.

We sincerely hope that our changes are to your satisfaction.

Yours sincerely

Louise Faurholt Øbro

Answer to reviewer 1:

Thank you for your time and your positive feedback; your comments have helped improving the clarity of the study.

  • you had an extremely small participant cohort to report against 
  • Thank you for this comment. We know that it was a relatively small group of participants (4 nurses and 1 doctor), however, small data samples in qualitative research is quite common if the data is rich and detailed as in the present study.
  • there is no report/limitation regarding the difference in coaching training the nurses received, which may well have impacted the findings and I feel needs recognition in the limitation.  

- As suggested, we have now described the limitation of the difference (page 13-14).

  • You have provided no recommendations regarding
  • future coaching training
  • Technology literacy or future digital health education requirement

- Thank you for the suggestions. We have now specified this in the discussion (page 13).

Reviewer 2 Report

Review Healthcare Professionals’ Experiences and Perspectives of Facilitating Self-Management Support via mHealth and Health  Coaching.

In general, I consider that the article needs to include important improvements and revisions. There is a need to clarify adequately what the objectives were, what was measured and how. It is important to detail the interventions made and to explain everything clearly.

The abstract does not mention anything in the first part about mhealth, i.e. because it is used? It is not adequately justified in the abstract.

Why is prostate cancer mentioned exclusively in the introduction but not in the title of the article or in the abstract? If it is a specific focus on cancer patients, it should be stated. If it is not directly related, other examples should be used in the introduction. 

In the Method section, the design of the training should be properly clarified. As it is currently explained, it is not properly understood. I understand that an initial 1-day training was given to group 1 (how many nurses participated in this group? Who defined the contents? was there a questionnaire or pre-test to identify their needs?) The second training was followed by group 2 on the basis of what was indicated by the nurses of group 1. Were they different nurses? Did the nurses of group 1 have access to the second day of group 2?

Why was the "Wheel of Life" tool used and not some other tool, and was it based on any theory or theoretical construct to make these decisions?

In the intervention sub-section, the procedure followed needs to be much more detailed. The text states that the nurse coaches in group 1 and the patients identified areas needing improvement. The interest in improving the nurse's ability to follow up the patient has been mentioned throughout. Here we talk about shortcomings detected by the patients, where did this information come from? How was it requested? If interventions between professionals and patients were carried out, the general objective should be modified and included in the introduction and abstract, as well as in the description of the sample.

The use of the application My Course (which has not been mentioned so far) is also not well explained, could it be better explained why it has been decided to use this tool and what role it plays in the training of nurses or in the subsequent coaching? 

Shouldn't figure 1b be in results?

In the results section, what does it mean that the nurses were less distracted, were they before, and what was the cause?

I understand that in the coaching training nurses were not trained in the use of devices. In that case, it was expected that they did not feel comfortable with their use and did not understand what they were using it for. I am still not very clear about that either.

There is a small typo in line 227.

Given the small sample size, such phrases should be avoided:Most of the nurse coaches described.

The disadvantages encountered during the implementation of the technology are discussed in section 3.4. This should be seen more as a methodological flaw than as a result obtained. The nurses did not know how to handle this technology, it was included at the same time as the learning of the coaching and this could have biased and hindered the achievement of better results.

What exactly was the rationale for including music in this intervention? It should be clarified because I don't quite understand it.

As for the discussion, it tends to generalise results when this is not possible in qualitative research, especially with such a small sample size. Nothing is included about the participating doctor? How did he or she contribute to the study? Who interpreted the texts?

Author Response

Response to the Reviewers

Dear reviewers,

Thank you very much for your interest in our manuscript. We are grateful that you have decided to give us the opportunity to revise the manuscript according to the reviewer’s comments to meet your expectations for publication. We hereby resubmit our revised manuscript.

We appreciate the reviewers’ useful comments and the editorial instructions. We have answered all comments below point by point and followed the requested editorial instructions.

We sincerely hope that our changes are to your satisfaction.

Yours sincerely

Louise Faurholt Øbro

Answer to reviewer 2:

Thank you for reading our manuscript and for your useful feedback.

  • In general, I consider that the article needs to include important improvements and revisions. There is a need to clarify adequately what the objectives were, what was measured and how. It is important to detail the interventions made and to explain everything clearly.

  • Thank you for the suggestion, the intervention is now described clearer and in more detail (page 2). Our study is a qualitative study, and hence the objective of this study was to explore the healthcare professionals' experiences and perspectives of the intervention.

  • The abstract does not mention anything in the first part about mhealth, i.e. because it is used? It is not adequately justified in the abstract.

  • We have mentioned mHealth in the introduction section in the abstract (see page 1) .

  • Why is prostate cancer mentioned exclusively in the introduction but not in the title of the article or in the abstract? If it is a specific focus on cancer patients, it should be stated. If it is not directly related, other examples should be used in the introduction.

  • Thank you for your comment. We have now included “patients with low-risk localized prostate cancer” in the title. Prostate cancer is further mentioned in the Method- and Result section(page 3-4 and 8)

  • In the Method section, the design of the training should be properly clarified. As it is currently explained, it is not properly understood. I understand that an initial 1-day training was given to group 1 (how many nurses participated in this group? Who defined the contents? was there a questionnaire or pre-test to identify their needs?) The second training was followed by group 2 on the basis of what was indicated by the nurses of group 1. Were they different nurses? Did the nurses of group 1 have access to the second day of group 2?
  • We recognize that the design of the training was unclear and that how we identified the needs could be more detailed. This is now clarified in the manuscript (page 3).

  • Why was the "Wheel of Life" tool used and not some other tool, and was it based on any theory or theoretical construct to make these decisions?

  • Thank you for the comment. It is now described more clearly (page 3-5).

  • In the intervention sub-section, the procedure followed needs to be much more detailed. The text states that the nurse coaches in group 1 and the patients identified areas needing improvement. The interest in improving the nurse's ability to follow up the patient has been mentioned throughout. Here we talk about shortcomings detected by the patients, where did this information come from? How was it requested? If interventions between professionals and patients were carried out, the general objective should be modified and included in the introduction and abstract, as well as in the description of the sample.

  • As suggested, we have described the procedure more detailed (page 3-5). There was no intervention between the healthcare professionals and patients, and we have therefore not modified the objective. Our study was a qualitative study, and it was the professionals that provided the intervention to the patients. The objective of this study was to explore the professionals’ experiences of providing the intervention.

  • The use of the application My Course (which has not been mentioned so far) is also not well explained, could it be better explained why it has been decided to use this tool and what role it plays in the training of nurses or in the subsequent coaching?

  • We appreciate your comment; and this is now explained more detailed (page 3)).

  • Shouldn't figure 1b be in results?

  • Figure 1b is an example of the tool in a “filled out” version, this is now made more clear in the text to avoid confusions.

  • In the results section, what does it mean that the nurses were less distracted, were they before, and what was the cause?

  • As suggested, this is now described in more detail (page 7-8).

  • I understand that in the coaching training nurses were not trained in the use of devices. In that case, it was expected that they did not feel comfortable with their use and did not understand what they were using it for. I am still not very clear about that either.

  • Thank you for addressing this. You are right; the nurses were not trained in using the devices. We have now elaborated on this in the discussion-section (page 14).

  • There is a small typo in line 227.

  • Thank you for addressing the typo, it is now corrected

  • Given the small sample size, such phrases should be avoided:Most of the nurse coaches described.

  • This is now corrected.

  • The disadvantages encountered during the implementation of the technology are discussed in section 3.4. This should be seen more as a methodological flaw than as a result obtained. The nurses did not know how to handle this technology, it was included at the same time as the learning of the coaching and this could have biased and hindered the achievement of better results.

  • Thank you for your comments on this. As recommended we have now elaborated on this in the discussion (page 13-14). The section 3.4 remains, as it is a result do to our explorative objective of the study.

  • What exactly was the rationale for including music in this intervention? It should be clarified because I don't quite understand it.

  • As suggested this is now clarified in the manuscriptin more detail (page 4).

  • As for the discussion, it tends to generalise results when this is not possible in qualitative research, especially with such a small sample size. Nothing is included about the participating doctor? How did he or she contribute to the study? Who interpreted the texts?
  • Thank you. As recommended, we have now reformulating the results section (page 7-12).

The doctor was a part of the focus group, see line 226. We cannot state which sections refers to the doctor, as it would undermine the anonymity.

Charlotte Handberg and Louise Faurholt Øbro interpreted the texts, which is described in the methods section and the Authors’ contributions, page 14

Reviewer 3 Report

The authors appropriately have properly outlined the method of achieving to study objective. Thus, the following suggestions were made for the improvement of the manuscript quality.

Abstract:

1.     The abstract word count is much more than the maximum word count (200 words or less). Therefore, it is necessary to summarize the abstract subsections, especially the sub-section of the introduction from the abstract section and it is sufficient to mention the importance of the topic.

2.     The time period of the study should be stated in the method subsection of the abstract section.

3.     At the end of the abstract section, there was no keyword section.

Introduction:

1.     A brief description of health coaching can help to better understand this concept for a wider range of readers of this study. Therefore, it is suggested that the concept of health coaching be briefly defined in the introduction section.

2.     It is necessary to provide more explanation concerning the importance of self-care management mobile apps in chronic diseases, especially in patients with cancer in the introduction section. It is suggested to use the article entitled “Development of a Mobile-Based Self-care Application for Patients with Breast Cancer-Related Lymphedema in Iran” in this regard. Available from:  https://pubmed.ncbi.nlm.nih.gov/36198310/

Methods:

1.     In the purposeful selection of samples, what criteria are included for selecting samples (nurses)? It is necessary to explain the criteria included in the setting and sampling subsection of the method section.

2.     Why have the authors expressed the sentence " Finally, we created a model representing the analytical findings in a hierarchy" with reference? Did the authors mean that the model created in the cited reference exists?

3.     Please correct the typos “manage-mentwas” error on page 8 line 227.

Results:

1.     I have no comment in the results section. The results are presented logically.

Discussion:

1.     The results of the present study have shown that the low ability or lack of ability to use mobile health in nurses has a negative effect on their coaching and being professional. Considering the role of nurses as the front-line caregivers in providing healthcare to patients and the vital role of mHealth in the self-care of patients with chronic diseases including cancer, improving the competence of nurses to use this technology to improve the quality of providing services to patients is even more necessary. The authors have only mentioned education as a solution to solve the challenge of using mHealth in lines 439-441. Studies also confirm that education and reducing the resistance of practitioners to accepting and using mHealth is effective. However, very important factors such as socio-technical, ergonomic, and human factors are effective in designing mobile-based applications and engaging users, which have not been addressed. See the discussion section of the following article: https://pubmed.ncbi.nlm.nih.gov/36198310/

Conclusion:

1.     Considering the focus of the study on facilitating self-management support via mHealth and health coaching, it is necessary to mention the conclusion related to the use of mHealth by nurses and patients in the conclusion section.

Author Response

Response to the Reviewers

Dear reviewers,

Thank you very much for your interest in our manuscript. We are grateful that you have decided to give us the opportunity to revise the manuscript according to the reviewer’s comments to meet your expectations for publication. We hereby resubmit our revised manuscript.

We appreciate the reviewers’ useful comments and the editorial instructions. We have answered all comments below point by point and followed the requested editorial instructions.

We sincerely hope that our changes are to your satisfaction.

Yours sincerely

Louise Faurholt Øbro

Answer to reviewer 3:

Thank you for your time and your valuable feedback.

Abstract:

  1. The abstract word count is much more than the maximum word count (200 words or less). Therefore, it is necessary to summarize the abstract subsections, especially the sub-section of the introduction from the abstract section and it is sufficient to mention the importance of the topic.

  • Thank you for drawing our attention to this. We have corrected the abstract, page 1.

  1. The time period of the study should be stated in the method subsection of the abstract section.

  • It is now stated in the abstract, page 1.

  1. At the end of the abstract section, there was no keyword section.

  • Thank you for your comment. Keywords is now added, page 1.

Introduction:

  1. A brief description of health coaching can help to better understand this concept for a wider range of readers of this study. Therefore, it is suggested that the concept of health coaching be briefly defined in the introduction section.

  • As suggested, this is now clarified in the introduction where we have added a description of health coaching (page 2).

  1. It is necessary to provide more explanation concerning the importance of self-care management mobile apps in chronic diseases, especially in patients with cancer in the introduction section. It is suggested to use the article entitled “Development of a Mobile-Based Self-care Application for Patients with Breast Cancer-Related Lymphedema in Iran” in this regard. Available from:  https://pubmed.ncbi.nlm.nih.gov/36198310/

  • Thank you for the suggestion and reference, it is now added in the introduction, page 2.

Methods:

  1. In the purposeful selection of samples, what criteria are included for selecting samples (nurses)? It is necessary to explain the criteria included in the setting and sampling subsection of the method section.

  • We appreciate this observation, and we have adjusted accordingly in the manuscript, page 3.

  1. Why have the authors expressed the sentence "Finally, we created a model representing the analytical findings in a hierarchy" with reference? Did the authors mean that the model created in the cited reference exists?

  • Thank you for your comment on this. The methodology Interpretive Description recommends the use of a model of the findings, which are the reason for adding a reference, page 7.

  1. Please correct the typos “manage-mentwas” error on page 8 line 227.

  • Thank you. This is now corrected.

Results:

  1. I have no comment in the results section. The results are presented logically.

  • Thank you for your positive feedback on this section

Discussion:

  1. The results of the present study have shown that the low ability or lack of ability to use mobile health in nurses has a negative effect on their coaching and being professional. Considering the role of nurses as the front-line caregivers in providing healthcare to patients and the vital role of mHealth in the self-care of patients with chronic diseases including cancer, improving the competence of nurses to use this technology to improve the quality of providing services to patients is even more necessary. The authors have only mentioned education as a solution to solve the challenge of using mHealth in lines 439-441. Studies also confirm that education and reducing the resistance of practitioners to accepting and using mHealth is effective. However, very important factors such as socio-technical, ergonomic, and human factors are effective in designing mobile-based applications and engaging users, which have not been addressed. See the discussion section of the following article: https://pubmed.ncbi.nlm.nih.gov/36198310/

  • We recognize that the mentioned areas are very important. We have added a description of these topics in the discussion section, page 13.

Conclusion:

Considering the focus of the study on facilitating self-management support via mHealth and health coaching, it is necessary to mention the conclusion related to the use of mHealth by nurses and patients in the conclusion section.

  • Thank you for this comment. We have specified this in the conclusion, page 14.

Round 2

Reviewer 2 Report

I thank the authors for all the modifications made.

I found a typo in line 146, I think they meant "see figure...".

I think the intervention could be explained a little better, I still have doubts after reading the document. However, it is much improved from the initial version. As an example, the part where increased water consumption is encouraged is explained, but the other interventions and their purpose are not detailed. Figure 3, which summarises the training, should include more detailed and descriptive information.

Author Response

Response to Reviewer

To reviewer 2,

Thank you for your useful comments. We hereby resubmit our revised manuscript.

We have answered the comments below point by point.

We sincerely hope that our changes are to your satisfaction.

Yours sincerely

Louise Faurholt Øbro

________________________________________________________________________________

Answer to reviewer 2:

  • I found a typo in line 146, I think they meant "see figure...".

Thank you for this comment. The typo is now corrected. 

  • I think the intervention could be explained a little better, I still have doubts after reading the document. However, it is much improved from the initial version. As an example, the part where increased water consumption is encouraged is explained, but the other interventions and their purpose are not detailed.

As suggested, the intervention and purpose is now described more detailed, see page 4. 

  • Figure 3, which summarises the training, should include more detailed and descriptive information.

Thank you for this comment. Figure 3 describes the sessions the patients attended throughout the 19 weeks intervention. This is corrected in the text, see page 4.